# 1,4-Disubstituted 1*H*-1,2,3-Triazoles for Renal Diseases: Studies of Viability, Anti-Inflammatory, and Antioxidant Activities

**DOI:** 10.3390/ijms21113823

**Published:** 2020-05-28

**Authors:** Ching-Yi Cheng, Ashanul Haque, Ming-Fa Hsieh, Syed Imran Hassan, Md. Serajul Haque Faizi, Necmi Dege, Muhammad S. Khan

**Affiliations:** 1Graduate Institute of Health Industry Technology, Research Center for Chinese Herbal Medicine and Research Center for Food and Cosmetic Safety, Chang Gung University of Science and Technology, No. 261, Wenhua 1st Rd., Guishan Dist., Taoyuan City 333, Taiwan; 2Department of Pulmonary Infection and Immunology, Chang Gung Memorial Hospital at Linkou, No. 5, Fuxing St., Guishan Dist., Taoyuan City 333, Taiwan; 3Department of Chemistry, College of Science, University of Hail, Ha’il 81451, Saudi Arabia; 4Department of Chemistry, College of Science, Sultan Qaboos University, P.O. Box 36, Al-Khod 123, Sultanate of Oman; imranhassan2@gmail.com (S.I.H.); msk@squ.edu.om (M.S.K.); 5Department of Biomedical Engineering, Chung Yuan Christian University, No. 200 Zhongbei Rd., Zhongli Dist., Taoyuan City 320, Taiwan; 6Department of Chemistry, Langat Singh College, B.R.A. Bihar University, Muzaffarpur, Bihar 842001, India; faizichemiitg@gmail.com; 7Department of Physics, Faculty of Arts and Sciences, Ondokuz Mayis University, 55139 Atakum, Turkey; necmid@omu.edu.tr

**Keywords:** tumor necrosis factor-α (TNF-α), cytosolic phospholipase A_2_ (cPLA_2_), prostaglandin E_2_ (PGE_2_), matrix metalloproteinase-9 (MMP-9), inducible nitric oxide synthase (iNOS)

## Abstract

Inflammation is a hallmark of many metabolic diseases. We previously showed that ferrocene-appended 1*H*-1,2,3-triazole hybrids inhibit nitric oxide (NO) production in in vitro models of lipopolysaccharide-induced inflammation in the BV-2 cell. In the present study, we explored the viability, anti-inflammatory, and antioxidant potential of ferrocene-1*H*-1,2,3-triazole hybrids using biochemical assays in rat mesangial cells (RMCs). We found that, among all the ferrocene-1*H*-1,2,3-triazole hybrids, **X2**–**X4** exhibited an antioxidant effect on mitochondrial free radicals. Among all the studied compounds, **X4** demonstrated the best anti-inflammatory effect on RMCs. These results were supplemented by in silico studies including molecular docking with human cytosolic phospholipase A_2_ (cPLA_2_) and cyclooxygenase 2 (COX-2) enzymes as well as absorption, distribution, metabolism, excretion, and toxicity (ADMET) profiling. Besides, two new crystal structures of the compounds have also been reported. In addition, combining the results from the inducible nitric oxide synthase (iNOS), cPLA_2_, COX-2, and matrix metalloproteinase-9 (MMP-9) enzymatic activity analysis and NO production also confirmed this argument. Overall, the results of this study will be a valuable addition to the growing body of work on biological activities of triazole-based compounds.

## 1. Introduction

Inflammation, the body’s immune response to stimulus, is a complex biological process/defense mechanism that is triggered by external or internal stimuli. It is associated with minor infections/injuries and with major disorders such as cancer, cystitis, arthritis, asthma, and psoriasis [1]. In renal disease, mesangial cells (MCs) play a vital role in the evolution of immune-mediated inflammation. It has been shown that tumor necrosis factor-α (TNF-α) levels are associated with the development of nephropathy [2]. Renal expression and circulating levels of bioactive TNF-α increase in lupus nephritis and correlate with disease activity [3]. TNF-α has been implicated in renal inflammation by its upregulation of inflammatory genes such as cyclooxygenase-2 (COX-2), cytosolic phospholipase A_2_ (cPLA_2_), prostaglandin E_2_ (PGE_2_), matrix metalloproteinase-9 (MMP-9), and inducible nitric oxide synthase (iNOS) [4,5]. Eicosanoids, which are involved in immune-mediated renal inflammatory diseases, are generated from arachidonic acid metabolism via cPLA_2_ and COX-2. We previously identified some traditional Chinese medicines that were effective in treating TNF-α-induced MC inflammation via downregulation of cPLA_2_ and COX-2 [6]. Several other studies also indicated the significance of MMPs and tissue inhibitors of metalloproteinases (TIMPs) in the progression of glomerulonephritis [7]. The production of nitric oxide (NO) is catalyzed by NO synthase (NOS), which degrades L-arginine to L-citrulline and NO. iNOS is one NOS isoform that is expressed by macrophages and other tissues in response to (pro) inflammatory mediators [8]. Therefore, the modulation of MC responses could offer a pathophysiology-based therapeutic approach to prevent glomerular injury.

To alleviate chronic inflammation and its related pain, the current drugs of choice are nonsteroidal anti-inflammatory drugs (NSAIDs) [8]. However, long-term use of classical NSAIDs has been associated with complications, including gastrointestinal ulcers, gastroduodenal erosions, renal dysfunction, and hepatotoxicity, mainly because of their variable selectivity towards COX enzymes [9]. These drawbacks led to the search for new synthetic anti-inflammatory agents, which has identified several new carbocyclic and heterocyclic small molecules that are in clinical use. For example, celecoxib, ramifenazone, and famprofazone are well-known examples of pyrazole-based anti-inflammatory drugs. However, serious side-effects, including bone marrow depression, water and salt retention, and carcinogenesis prompted researchers to continue the search for new anti-inflammatory drugs.

Among heterocyclic cores, 1,2,3-triazole enjoys a reputable position in the area of drug discovery. Easy synthesis/functionalization, ability to interact with bioreceptors, rigid, and stable structure are some of the leading reasons behind the popularity of this core. [10] Similarly, organometallic compounds such as ferrocene are known for their high metabolic stability, redox behavior, lipophilic and non-toxic features. Recently, Guo and co-workers demonstrated that ferrocene-based compounds could significantly reduce the LPS-induced NO secretion, IL-6, and TNF-α levels [11] The structure activity relationship (SAR) studies indicated that the presence of planar spacers attached to the ferrocene fragment is beneficial for the inhibition of NO production in the in vitro model. It has been suggested that the new classes of anti-malarial, anti-tubercular, and anti-microbial agents could be achieved by merging these two cores, i.e., ferrocene-triazole hybrids [12]. Previously, we reported that 1,4-disubstituted 1*H*-1,2,3-triazoles (**F**, **X1**–**X5**, Scheme 1) inhibit NO production in in vitro models of lipopolysaccharide (LPS)-induced inflammation [9]. Prompted by these results, we decided to use rat mesangial cells (RMCs) as an in vitro cell culture model to examine the effects of ferrocene-1*H*-1,2,3-triazole hybrids on TNF-α-induced inflammation. First, the cell viability of these compounds on RMCs were evaluated. Next, we explored the effect of these compounds on TNF-α-induced inflammation in vitro. In addition, we report two new crystal structures.

## 2. Results and Discussion

### 2.1. Synthesis and Characterization

We previously reported the detailed synthesis, structural and electrochemical characterization of **X1**–**X5** [9]. Briefly, these compounds were obtained in good yields and purity by azide-alkyne chemistry using ethynyl ferrocene and substituted amines. We were able to grow two new crystals (**X1** and **X5**) and the results are discussed below.

#### X-ray Crystal Structure

Single crystals of **X1** and **X5** were collected from dichloromethane solution after slow evaporation. The experimental details and the crystal data for **X1** and **X5** are given in Table 1, while their molecular structures are shown in Figure 1a,b. The structural parameters for both ferrocenyl substituents in compounds **X1** and **X5** were within the normal ranges, and the iron atom is sandwiched almost perfectly centrally between the two cyclopentadienyl rings in compound **X1** compared with **X5**. The distance of iron from the two cyclopentadienyl rings differs slightly in the two compounds. The iron was observed at 1.649 (5) and 1.652 (5) Å from the plane defined by the two cyclopentadienyl systems in **X1** but at 1.642 (5) and 1.652 (5) Å in compound **X5**. In both structures, the ferrocene has adopted an eclipsed conformation. The NO_2_ group attached to the benzene ring formed dihedral angles of 5.160 (7)° in compound **X1** and 61.41 (7)° in compound **X5**. The variation in the dihedral angle may be associated with *ortho*- and *para*-substitution of the NO_2_ group in compounds **X1** and **X5**, respectively. Single-crystal data indicate nitro N–O bond lengths of 1.216 (5) and 1.212 (5) Å in compound **X1** and 1.205 (5) and 1.216 (5) Å in compound **X5**, respectively. The crystal data indicate an N–O double bond and involvement of greater resonance in the *para*-containing nitro group in **X1** compared with the *ortho*-substitution in compound **X5**.

As is clear from the structure (Figure 1), the benzene ring (C13–C18) and cyclopentadienyl ring (C6–C10) form dihedral angles to the central triazole ring (N1–N3/C11–C12) at 17.09 (8)° and 12.58 (8)°, respectively, in compound **X1**. However, the benzene ring (C13–C18) and cyclopentadienyl ring (C6–C10) form dihedral angles to the central triazole ring (N1–N3/C11–C12) at 9.56 (7)° and 34.83 (7)°, respectively, in compound **X5**. This variation is due to the presence of substituted nitro groups at different positions in these compounds. The details of the intermolecular hydrogen bonds present in both compounds **X1** and **X5** are summarized in Table 2. Compound **X1** forms a dimer because of C—H⋯O interactions (Figure 2a) and is stabilized by packing because of *π*⋯*π* interactions (Figure 2b). Compound **X5** forms a one-dimensional polymeric structure because of C—H⋯N interactions (Figure 2c) and is also stabilized by packing due to *π*⋯*π* interactions (Figure 2d). The presence of such noncovalent interactions is very important in biological applications to allow understanding of many biological processes [13].

### 2.2. In Vitro Characterizations

#### 2.2.1. Antioxidant Effect on RMCs

To identify a non-toxic dose, we first examined the cell viability on RMCs of various concentrations of ferrocene (**F**) and ferrocene-1*H*-1,2,3-triazole hybrids **X1**–**X5** using the 3-(4,5-dimethylthiazol-2-yl)-2,5-diphenyltetrazolium bromide (MTT) assay. As shown in Figure 3a, the compounds were noncytotoxic at concentrations up to 50 μg/mL. It is worthwhile to mention here that the incorporation of triazole fragment to the ferrocene did not affect the cell viability profile of the ferrocene. Since all the hybrid compounds exhibited similar cell viability effects on RMCs, it can be concluded that merging these two pharmaceutically active cores is a safe strategy. Accordingly, a concentration of 12.5 µg/mL ferrocene-1*H*-1,2,3-triazole hybrids was used for all subsequent experiments.

Free radicals have the main role in inflammation and are divided into two categories, including reactive oxygen species (ROS) and reactive nitrogen species (RNS). ROS contains superoxide (O_2_^−^), hydroxy radical (^●^OH), peroxy radical (ROO^●^) and H_2_O_2_. Superoxide (O_2_^−^) is mainly produced in the mitochondrial electron transfer chain. When the cell performs respiration to generate ATP, about 1%–3% of the electrons leak out in a series of transmission processes and combine with oxygen to produce superoxide [14]. In addition, cells will also produce superoxide through nicotine adenine dinucleotide phosphate (NAD (P) H) oxidase. Next, to quantitate the free radicals (ROS) in cells and in the mitochondria, we performed dihydroethidium (DHE) and MitoSOX staining (mitochondrial staining) followed by flow cytometric analysis and immunofluorescence, respectively. The results of this study indicated that, except hybrids **X1** and **X5** (Figure 3b), all other compounds showed antioxidant activities. Though there is no exact explanation behind the ineffectiveness of **X1** and **X5**, the antioxidant behavior of other hybrids (**X2**, **X3** & **X4**) and reference (**F**) compounds could be attributed to the following: Firstly, ferrocene-1*H*-1,2,3-triazole hybrid compounds are electroactive and can undergo facile redox changes. This means they have a propensity to scavenge free radicals (or at least alter their level). The electron-donating capability of the hybrid compounds is largely governed by the core attached to the ferrocene ring. For instance, electrochemical studies suggested that compound **X4** (E_ox_ (onset) = 0.32 V) has more tendency to get oxidized than **X3** (E_ox_ (onset) = 0.34 V) and **X1** (E_ox_ (onset) = 0.39 V) [9]. Interestingly, the same trend has been reflected in the antioxidant assay (antioxidant effect = **X4** > **X3** > **X1**). Secondly, it has been demonstrated that 1,2,3-triazole can be readily oxidized at the N1 position of the triazole core [15], thus provide an additional site for radical scavenging by such hybrid compounds. Of the antioxidant compounds, **X2**–**X4** exhibited a significant antioxidant effect on mitochondrial free radicals (Figure 3c). Absorption, distribution, metabolism excretion, and toxicity (ADMET) calculations (vide infra) supported this observation too, i.e., mitochondrial localization of the compounds. However, all the compounds cannot induce antioxidant enzymes, including HO-1 or SOD-2 (Appendix A). Therefore, we suggest that the antioxidant mechanisms of these compounds were not via antioxidant enzyme expression. In the future, we can further explore the antioxidant mechanisms of these compounds.

#### 2.2.2. Effect on iNOS Expression and NO Production in RMCs

RNS contains nitric oxide (NO^•^) and peroxynitroso (ONOO^−^) anions. Nitric oxide is a gas with unpaired electrons produced by NOSs in tissues, which is widely distributed in the body and has many physiological functions, defense, signal transduction, etc. The peroxynitroso anion is produced by the reaction of NO^•^ and superoxide. It is a kind of free radical with extremely active and super destructive power [16]. It has been shown that the progression of renal disease is associated with an increase in NO production and iNOS activity [17]. However, NO is a very polar molecule that is easily oxidized, whereas nitrite is a stable oxidative metabolite of NO. Therefore, NO production was represented by the accumulation of nitrite. In the present study, we measured nitrite levels and iNOS expression by Griess assay and real-time polymerase chain reaction (PCR), respectively. As shown in Figure 4a, a 24-h treatment with TNF-α (10 ng/mL) led to a 34.79-times higher expression of iNOS compared with control. Upon addition of ferrocene-1*H*-1,2,3-triazole hybrids (except **X5**) to the culture of RMCs, the iNOS level decreased remarkably compared with TNF-α-treated RMCs (*p* < 0.01). In a parallel experiment (Figure 4b), the production of NO was significantly increased in TNF-α-stimulated cells. Pretreatment with **X1**–**X5** decreased this TNF-α-induced response. Inhibition of NO and iNOS indicated that ferrocene-1*H*-1,2,3-triazole hybrids could protect against inflammation in RMCs.

To investigate whether ferrocene-1*H*-1,2,3-triazole hybrids could suppress iNOS gene expression in RMCs, the cells were treated with these compounds at a non-toxic dose of 12.5 μg/mL for 24 h and analyzed by real-time PCR. As shown in Figure 4c, compounds **F** and **X1**–**X5** did not spontaneously induce iNOS gene expression in RMCs. Among them, **X2** and **X4** attenuated endogenous iNOS expression, but only **F**, **X1**, and **X3** significantly cause this inhibition (Figure 4c). Compared with other compounds, **X5** was the weakest in inhibiting endogenous iNOS and did not reach statistical significance. The expression of endogenous iNOS in **X5** alone treatment is close to that in the control group (Figure 4c). It may explain why it did not inhibit the expression of iNOS under the stimulation of TNF-α (Figure 4a). From the results obtained, it seems that the triazole core has an important role in the suppression of iNOS expression. In compounds **X2** and **X5**, azole fragment is blocked by the substituents present at the *ortho* position of the ring attached to azole (see Figure 1b). In **X1** and **X3**, the substituents are present on the *para* position and did not block azole fragments (see Figure 1a), thus causing inhibition. This steric effect fact is also supplemented by a comparative low inhibition profile of ferrocene alone, e.g., without steric effect. In addition, we speculate that the structure of these compounds contains the iron in ferrocene to interact with NO induced by TNF-α, leading to decrease nitrite formation (Figure 4b). This may explain why **X5** does not inhibit TNF-α-induced iNOS expression but inhibits nitrite production. The ability of **F**, **X1**, and **X3** to suppress the NO concentration could be also ascribed to a number of factors, but we envision that the molecular weight (MW), hydrogen bonding capability, solubility profile, topology and planarity of the compounds played a major role here. Overall, these results implied that compounds **F**, **X1**, and **X3** were better at preventing the elevation of NO concentrations.

#### 2.2.3. Effect on the Expression of Inflammatory Proteins in RMCs

TNF-α is a pro-inflammatory cytokine that plays a vital role in human and experimental glomerulonephritis and lupus nephritis [3,4]. TNF-α has been reported to up-regulate inflammatory genes in various cells. Both of COX-2 and cPLA_2_ can be considered as indicators of inflammation and play a vital role in several renal inflammatory diseases. In our previous study, we have established the relationship between COX-2 and cPLA_2_ and suggested that TNF-α enhances PGE_2_ generation via cPLA_2_/COX-2 upregulation in RMCs [4]. In addition, increasing studies have reported the significance of MMPs and TIMPs in the progression of glomerulonephritis. MMP-9 produced by neutrophils participates in the progression of renal fibrosis [18]. Therefore, we investigate whether ferrocene-1*H*-1,2,3-triazole hybrids regulate the expression of inflammatory proteins in RMCs, the protein levels of COX-2, cPLA_2_, and MMP-9 were examined. As demonstrated in Figure 5a and Figure 6a, treatment with all ferrocene-1*H*-1,2,3-triazole hybrids significantly (*p* < 0.05) decreased the expression of all these proteins in RMCs in response to TNF-α treatment. The levels of transcription of COX-2 (Figure 5b) and MMP-9 (Figure 6b) were also examined. Among ferrocene-1*H*-1,2,3-triazole hybrids, **X3** and **X4** caused more significant inhibition of TNF-α-induced COX-2 mRNA level than **F** (Figure 5b), while **X2**, **X4**, and **X5** significantly attenuated more TNF-α-induced MMP-9 mRNA level than **F** (Figure 6b). These results indicate that combining ferrocene and triazole motifs is an intriguing strategy to achieve an enhanced anti-inflammatory effect. In all the cases, we observed that the ferrocene-1*H*-1,2,3-triazole hybrids are more active than the ferrocene alone. However, the cells-treated with **X3** spontaneously increased COX-2 protein expression (Figure 5c) and the cells treated with **X5** spontaneously increased MMP-9 transcription (Figure 6c). In addition, RMCs treated with **X2** do not decrease COX-2 mRNA level in response to TNF-α (Figure 5b). Taken together, among all the ferrocene-1*H*-1,2,3-triazole hybrids, **X4** demonstrated the best anti-inflammatory effect on RMCs. From the results of Figure 5a,b, we can speculate that the main pharmacological mechanism of **F**, **X1**, and **X2** may be to inhibit TNF-α-induced COX-2 protein synthesis rather than mRNA transcription inhibition. In contrast, the **X3**–**X5** group may inhibit COX-2 mRNA transcription hence protein levels. We further analyze the effect of these compounds on the activity of COX-2 and cPLA_2_. As shown in Figure 5d, treatment with all ferrocene-1*H*-1,2,3-triazole hybrids significantly decreased the activity of cPLA_2_ as well as COX-2 in RMCs in response to TNF-α treatment. In this study, we also proved that cPLA_2_ affects COX-2 activity by AACOCF_3_ (a cPLA_2_ inhibitor) pretreatment, which is consistent with our previous publication [4]. On the inhibitory mechanisms of ferrocene-1*H*-1,2,3-triazole hybrids in TNF-α-induced MMP-9 expression, **X2** and **X4** mainly inhibit transcription level, but the rest of the compounds may affect enzyme activity (Figure 6a,b). It is worth noting that the treatment of healthy cells with **X3** and **X5** under a safe dose will cause side effects, because these two compounds cause the expression of inflammatory proteins. It is not recommended to use it in healthy food or for preventing inflammation in the future.

### 2.3. In Silico Studies

#### 2.3.1. ADMET Predictions

To obtain more information on the bioactivity and potential use of the reported molecules as drugs, we predicted their ADMET; the results of this analysis are presented in Table 3. From the data, it is clear that the compounds possess acceptable physicochemical, pharmacokinetic, and toxicity profiles. For example, all compounds returned positive results for blood–brain barrier (BBB) transition, gastrointestinal (GI) absorption, and oral bioavailability criteria, indicating their ability to pass through the BBB and be absorbed into tissues. Compounds **X1**, **X2**, and **X5** exhibited similar GI absorption (*p* = 0.8939), while compound **X3** (*p* = 0.8706) and compound **X4** (*p* = 0.8252) had comparatively lower probability of absorption. The fact that all compounds showed category III acute oral toxicity indicates that they may be acceptable for oral delivery. Moreover, all compounds showed the probability of mitochondrial localization, which is well reflected in the results of the antioxidant assays (Figure 3c). It should be noted that all these values are calculated and therefore, in vivo studies are required to confirm these observations.

#### 2.3.2. Docking Results

Molecular docking studies have become a common technique to identify molecular targets for treating disease. Using this method, one can easily predict the biological potential of compound(s), and active/catalytic sites within an enzyme. Biological results indicated that ferrocene-1*H*-1,2,3-triazole hybrids **X1**–**X5** exhibit neuroprotective effects via the inhibition of NO production in microglial cells (BV-2). To rationalize this observation at the supramolecular level, we carried out docking studies of the compounds with the enzymes established for participation in inflammatory processes. It is reported that compounds bearing 1,2,3-triazole core exhibit an anti-inflammatory effect via the inactivation of microglia localized COX isoenzymes [19]. Similarly, Chuang et al. [20] have shown that cPLA_2_ plays a crucial role in ROS/NO signaling in LPS activated BV2 cells; thus, cPLA_2_ can be considered as an intriguing therapeutic target for inflammation control.

In the present study, we carried out shape-based docking studies of compounds into the active sites of cPLA_2_ (apo form, PDB code: 1CJY) [21] and COX-2 (PDB code: 6COX) [22] using Autodock vina tools [23]. Since the crystal structure of cPLA_2_ had some missing regions, it was modelled using an online server, SWISS Model prior to the docking studies [24]. The results of the docking studies indicated that all hybrid compounds preferred to enter the active site of the enzyme and interact with the nearby residues (Figure 7). For instance, it has been reported that, in order to impart COX-2 inhibitory activity, a ligand should interact with conserved (His90), nonconserved (Arg513) and nearby residues (Arg120, Tyr355 and Glu524, Trp387, Phe518, Ser530, Arg120, Tyr355, Glu524 and Val523) of COX-2 via polar and nonpolar interactions [13]. Our docking result indicated that the ferrocene-1*H*-1,2,3-triazole hybrids prefer to enter the active site of the target and interact via multiple H-bonds (Table 4).

Docking studies with cPLA_2_ (PDB code: 1CJY) indicated that, except **X2**, compounds **X1**–**X5** did not make any contact with the catalytic residue Ser228 or with the “oxyanion hole” (residues Gly197/Gly198) of the enzyme [25]. However, they interacted with the nearby residues of the catalytic site of cPLA_2_ (Figure 8 and compiled in Table 4). The binding propensity of the ligand is significantly controlled by the functionalities present over the phenyl ring of triazole fragment. Unfortunately, we found that the docking energies do not match with the observed activity. However, in both cases, we observed that compound **X2** has the highest binding energies than the others, which is not very well reflected in in vitro studies. This can be ascribed to several factors, but we envisaged that high molecular weight (MW), hydrogen bonding propensity, solubility, and structural rigidity played important roles here.

For instance, compound **X2** has five HBA units (due to two nitro groups), which explains why it showed high affinity and binding energy with the enzymes in docking studies. At the same time, compound **X2**, due to its comparatively higher MW (Table 3), lower solubility (log*P* = 5.54, Table 3), and less structural rigidity (due to the presence of rotatable biphenyl units) leads to lower in vitro activity. Similarly, compounds **X3** and **X4**, bearing halogen atoms over the phenyl ring with moderate MW and lowest number of HBA units showed high activities.

## 3. Materials and Methods

### 3.1. Synthesis and Characterization

All chemicals, except where stated otherwise, were obtained from Merck (Darmstadt, Germany) and used as received. The detailed synthesis and characterization of compounds 1-(4-nitrophenyl)-4-ferrocenyl-1*H*-1,2,3-triazole (**X1**), 1-(4,4′-dinitro-2-biphenyl)-4-ferrocenyl-1*H*-1,2,3-triazole (**X2**), 1-(3-chloro-4-fluorophenyl)-4-ferrocenyl-1*H*-1,2,3-triazole (**X3**), 1-(4-bromophenyl)-4-ferrocenyl-1*H*-1,2,3-triazole (**X4**), and 1-(2-nitrophenyl)-4-ferrocenyl-1*H*-1,2,3-triazole (**X5**) can be found in our previous report [9]. Single-crystal X-ray diffraction for compounds **X1** and **X5** was carried out on a Stoe IPDS 2 diffractometer equipped with a graphite crystal monochromator situated in the incident beam for data collection at room temperature [26]. The determination of unit cell parameters and data collection were performed using Mo-Ka radiation (λ = 0.71073 Å). [27] Unit cell dimensions were obtained with least-squares refinements, and all structures were solved by direct methods with SHELXT2015 [28,29]. All H atoms were located from difference-Fourier maps, but in the final cycles of refinement they were included in the calculated positions and treated as riding atoms: C-H = 0.93–0.98 Å with Uiso (H) = 1.2 Ueq (C). Selected geometric parameters (Å, º) of the compounds are listed in Appendix A.

### 3.2. In Vitro Characterizations of Synthesized Compounds

#### 3.2.1. Cell Culture in the Presence of Ferrocene-1*H*-1,2,3-Triazole Hybrids

RMCs (from the American Type Culture Collection (ATCC; Rockville, MD, USA)) were cultured in Dulbecco’s minimal essential medium containing 10% heat-inactivated fetal bovine serum (FBS) at 37 °C in a 5% CO_2_ atmosphere. For the measurement of protein expression, enzymatic activity, and mRNA levels, 2 × 10^5^ cells/well were seeded into 12-well plates. In anti-inflammation experiments, cells were starved in 0% FBS and cultured with or without **X1**–**X5** for 2 h prior to TNF-α (R&D Systems, Minneapolis, MN, USA) treatment for 24 h.

#### 3.2.2. Viability Assay

Cells were seeded at a density of 3 × 10^5^ cells per well into 96-well plates and treated with different concentrations of **X1**–**X5** (from 1.5625 to 100 μg/mL in 1% DMSO) for 24 h, followed by the addition of 0.5 mg/mL MTT (Merck) for another 2 h. The MTT solution was then discarded and 100 μL of DMSO (Merck) was added to dissolve the formazan crystals. The level of colored formazan was analyzed on a microplate reader (SpectraMax 250, Molecular Devices, San Jose, CA, USA) at a wavelength of 540 nm. The values were determined by comparing the optical density of the **X1**–**X5**-treated group with that of the vehicle-treated group (1% DMSO).

#### 3.2.3. Gelatin Zymography

MMP-9 expression was analyzed as previously described [30]. Briefly, after treatment, the culture medium was collected and centrifuged at 10,000× *g* for 5 min at 4 °C to remove cell debris. Next, the supernatants were mixed with 5× nonreducing sample buffer and electrophoresed on a 10% polyacrylamide gel containing 0.15% gelatin. After electrophoresis, the gel was washed twice in 2.5% Triton X-100 and then incubated in developing buffer at 37 °C overnight. After incubation, the gel was stained with staining buffer (30% methanol, 10% acetic acid, and 0.5% (wt/vol) Coomassie brilliant blue). Gelatinolytic activity was observed as white bands on a blue background.

#### 3.2.4. Nitrite Production

After cell treatment, the conditioned medium was collected and analyzed for nitrite production by Griess assay [31]. Briefly, 50 μL of a solution containing 4% sulfanilic acid, 0.2% N-(1-naphthyl) ethylenediamine dihydrochloride, and 10% phosphoric acid was added to 50 μL of conditioned medium. Absorbance reading was taken at 550 nm after samples were incubated at 25 °C for 10 min in the dark. The standard curve of various NaNO_2_ concentrations was used to calculate the NO production in the sample.

#### 3.2.5. Western Blotting for Inflammatory Proteins

Western blotting was conducted as previously described [4]. Briefly, after the cell culture, RMCs were washed, scraped, collected, and centrifuged at 45,000× *g* for 1 h at 4 °C to yield the whole-cell extract. The whole-cell extracts were quantitated, adjusted for concentration, denatured, resolved with 10% sodium dodecyl sulfate polyacrylamide gel electrophoresis, and transferred to polyvinylidene fluoride membranes (Millipore, Bedford, MA, USA). Membranes were incubated with primary anti-COX-2 (Cell Signaling Technology, Danvers, MA, USA), anti-cPLA_2_ (Cell Signaling Technology) or anti-GAPDH antibody, which was used as a loading control (Santa Cruz Biotechnology, Santa Cruz, CA, USA) at 4 °C overnight and secondary anti-rabbit or anti-mouse horseradish peroxidase antibody for 1 h at room temperature. The immunoreactive signals detected by enhanced chemiluminescence reagents were developed using a LumiFlash Ultima chemiluminescent substrate horseradish peroxidase system (Visual Protein Biotech Corporation, Taipei, Taiwan). The densitometry units of COX-2, cPLA_2_ and GAPDH were quantified by ImageLab TM 5.0 Software (Bio-Rad Laboratories, Inc., Hercules, CA, USA).

#### 3.2.6. Real-time PCR

Real-time PCR was performed using the CFX Connect Real-time PCR Detection System (Bio-Rad Laboratories) to determine the expression of inflammatory genes, as per the workflow steps described previously [4]. Briefly, total RNA was extracted and provided as a template for cDNA reverse transcription. The thermal conditions used were 3 min at 95 °C, 40 cycles of 10 s at 95 °C and 30 s at 58 °C. Relative gene expression was determined by the 2^−^^ΔΔCt^ method. Gene expression was normalized relative to unstimulated cells and fold variation was normalized to levels of β-actin expression (an endogenous control). The primers used for real-time PCR were as follows: 5′–CGTGAAAAGATGACCCAGATCA–3′ (forward) and 5′–CTCCGG AGTCCATCACAATG–3′ (reverse) for β-actin; 5′–ACATTCAGGCAGCAGAGGA–3′ (forward) and 5′–CCACCACAGGCACAT CAC–3′ (reverse) for cPLA_2_, and 5′–CAAGAATCAAATTACC GCTGAAG–3′ (forward) and 5′–CGAAGGAAGGGAATGTTGTT–3′ (reverse) for COX-2; 5′–CGCTTTCACCAAGACTGTGA–3′ (forward) and 5′–GCATCCCAAGTACGAGTGGT–3′ (reverse) for iNOS.

#### 3.2.7. cPLA_2_ and COX-2 Enzyme Activity

The experimental procedure of COX-2 activity detection is in accordance with the manufacturer’s instructions (Abcam, Cambridge, UK). The following are simple instructions: First, we established a standard curve by a series of diluted resorufin standards. Resorufin is a redox fluorescent probe that can be used to visualize cell respiration directly. Next, for sample preparation, 5 × 10^6^ cells treated without or with compounds in the presence of TNF-α were scraped off from the culture plate and washed with cold PBS. After centrifuging at 500× g for 3 min, we discarded the supernatant and resuspended the cell pellet in 0.2 mL of lysis buffer with protease inhibitor cocktail on ice for 5 min. After centrifuging at 12,000× g 4 °C for 3 min, we collected the supernatant used as the sample. Sample and COX-2 positive control were separately loaded into each of the 96 wells at 20 μL and then 2 μL DMSO (for total COX activity detection) and 2 μL COX-2 inhibitor (Celecoxib, for COX-2 activity detection) were added in two groups. Reaction mix reagents (containing COX Probe, Diluted COX Cofactor and COX Assay Buffer) at 68 μL were loaded into each well and mixed enough, using a multichannel pipette to add 10 µL diluted arachidonic acid/NaOH solution into each well to initiate the reaction at the same time. After addition of the arachidonic acid, the fluorescence (Ex/Em = 535/587 nm) was measured immediately in a kinetic mode once every 15 sec for at least 30 min. One Unit COX activity = amount of COX which generates 1.0 µmol of resorufin per min.

The activity of COX-2 in the test samples is calculated as:(1)COX Activity=(BΔT × M)=pmol/min/mg or µU/mg
where

B = Amount of resorufin from Standard Curve (pmol).

ΔT = Reaction time (min).

M = Protein amount added into the reaction well (mg)

at pH 8.0, 25 °C.

Use the ΔRFU_535/587nm_ to obtain B pmol of resorufin generated by COX-2 during the reaction time (ΔT = T_2_ – T_1_).
∆RFU_535/587nm_ = (RFU_S2_ – RFU_S1_) – (RFU_I2_ – RFU_I1_)(2)
where

RFU_S2_ is the sample (DMSO) reading at time T2.

RFU_S1_ is the sample (DMSO) reading at time T1.

RFU_I2_ is the inhibitor sample (Celecoxib) at time T2.

RFU_I1_ is the inhibitor sample (Celecoxib) at time T1.

PLA_2_ sample that can utilize arachidonoyl thio-PC as a substrate can be measured by colorimetric cPLA_2_ assay kit (Abcam). Among them, any residual sPLA2 can be removed from the samples by using a membrane filter with a molecular weight cut-off of 30,000. To avoid any measurement of iPLA2 activity in the sample, use the iPLA2-specific inhibitor Bromoenol Lactone. Briefly, we added 15 µL of assay buffer (non-enzymatic control), positive control (bee venom PLA2) and sample (including iPLA2-specific inhibitor) to wells. We initiated the reactions for 1 h at room temperature by adding 200 µL substrate solution. The addition of 10 µL of DTNB/EGTA to each well stopped enzyme catalysis and developed the reaction for 5 min at room temperature. The absorbance was read at 414 nm using a microplate reader (SpectraMax 250). The activity of cPLA_2_ in the test samples is calculated as:(3)$${cPLA}_{2}Activity=\frac{\Delta A414/\min}{10.66\;\mathrm{mM}-1}\;\times \;\frac{0.225\;\mathrm{mL}}{0.01\;\mathrm{mL}}\;\times \;Sample\;dilution=µ\mathrm{mol}/\min/mL$$
where

A414/min = A414 (sample)−A414 (blank)/60 min

ΔA414/min = (A414/min_Sample_ − A414/min_Inhibitor_)

### 3.3. In Silico Studies

The physicochemical properties and oral bioavailability of compounds **X1**–**X5** were predicted using admetSAR 2 webserver. [32,33]. SMILE formats (as input) of the molecules were generated using Marvin 16.11.28.0, 2016, ChemAxon (http://www.chemaxon.com). Shape-based molecular docking studies were performed on an Intel (R) Core (TM) i5 CPU (2.3 GHz) with a Windows 2010-based operating system. Cif files of the ligands were converted to PDB files which were further used for docking analysis. The ligands were docked into the active sites of human cPLA_2_ (PDB code 1CJY) and COX-2 (PDB code: 6COX). The crystal structures were downloaded from the Brookhaven Protein Data Bank (http://www.rcsb.org). The pdbqt format of the compound and enzyme were obtained using AutoDock Tools (ADT) 1.5.4 [34]. Preparation of parameter files for grid and docking was done using the following parameters: grid box size of 108 × 86 × 78 Å with 0.375 Å spacing that included the whole enzyme. Autodock vina was used for the docking studies [35]. The docking results were analyzed using PyMol [36] for possible polar and hydrophobic interactions. Of the different conformations obtained, the least energetic and most stable conformation was selected.

## 4. Conclusions

A series of ferrocene-1*H*-1,2,3-triazole hybrids were investigated for their viability, anti-inflammatory, and antioxidant effects. X-ray single-crystal structure studies indicated that the position of the nitro group (*ortho* or *para*) controlled the structure of the compound (dimeric or polymeric) in the solid state. We found that, among all the ferrocene-1*H*-1,2,3-triazole hybrids, **X2**–**X4** exhibited antioxidant effect on mitochondrial free radicals, and among all, **X4** demonstrated the best anti-inflammatory effect on RMCs. In silico studies confirmed the safety of the compounds and their ability to bind to the active site of the pro-inflammatory factors. Overall, the results of the present study indicated that ferrocene-1*H*-1,2,3-triazole hybrid, **X4**, can be used as a lead to optimize and develop a new anti-inflammatory and antioxidant agent.

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
