# Peer review of "1,4-Disubstituted 1H-1,2,3-Triazoles for Renal Diseases: Studies of Viability, Anti-Inflammatory, and Antioxidant Activities"

_ijms, 2020, doi:10.3390/ijms21113823_

Round 1

Reviewer 1 Report

Authors performed cytotoxicity, antiinflammatory and antioxidant activities for the ferrocene compounds (X1-X5) they have previously reported synthesis.Throughout all experiments, ferrocene was used as a standard or comparison to X1-X5. However there is no mention why ferrocene was significant in this study. Role of ferrocene should be mentioned both in the design of structure and in biological assays.  

Author Response

Reply to reviewer’s comments

Journal:                   International Journal of Molecular Sciences 

Manuscript ID:       ijms-772595

Manuscript Title:    1,4‐Disubstituted 1H-1,2,3‐Triazoles for Renal Diseases: Studies of Viability, Anti-Inflammatory, and Antioxidant Activities

Authors:                  Ching-Yi Cheng*, Ashanul Haque*, Ming-Fa Hsieh*, Syed Imran Hassan, Serajul Haque Faizi, Necmi Dege and Muhammad S. Khan

We thank the editor and reviewers for their thorough review and for giving us an opportunity to revise our article. The comments/queries suggested by the editor and reviewers are replied and corrected in the revised manuscript (highlighted in red). We hope, the revised manuscript is suitable for publication in the International Journal of Molecular Sciences.

With best personal regards.

Sincerely yours,

Ching-Yi Cheng, Ph.D. (Corresponding Author)

Graduate Institute of Health Industry Technology, Research Center for Chinese Herbal Medicine and Research Center for Food and Cosmetic Safety, Chang Gung University of Science and Technology, No.261, Wenhua 1st Rd., Guishan Dist., Taoyuan City 33303, Taiwan.

Phone: 886-3-2118999, ext 5114

Fax: 886-3-2118866

Comment 1: Authors performed cytotoxicity, anti-inflammatory and antioxidant activities for the ferrocene compounds (X1-X5) they have previously reported synthesis. Throughout all experiments, ferrocene was used as a standard or comparison to X1-X5. However, there is no mention why ferrocene was significant in this study. Role of ferrocene should be mentioned both in the design of structure and in biological assays.

Reply: We thank the reviewer for the suggestion. In the design of the novel compounds in the study, the ferrocene (F) is used as a basic building block. The motivation behind comparing designed compounds (X1-X5) with F was to understand the additional effect of triazole. A paragraph focusing on the anti-inflammation function of azoles has been included in the revised MS. “Introduction” and “Results and Discussion” have also been updated.

See page: Page 2, Introduction 1, lines 68-79.

See page: Page 7, subsection 2.2.3, lines 215-235.

Reviewer 2 Report

In this manuscript the authors present the extension of their research about the 1,4‐disubstituted 1H-1,2,3‐triazoles compounds. Also, the authors synthesized and characterized new compounds and demonstrate the biological activity.    

The compounds present an acceptable physicochemical, pharmacokinetic, and toxicity profiles and the anti-inflammatory effect and antioxidant activity was demonstrated in vivo.

The experiment has been systematically performed and the manuscript is well organized therefore I suggest that this manuscript can be published on this journal after the correction of the following issues:

  • The compounds F, X1 and X3 were better for preventing the increase of NO concentrations. Please explain why.
  • The quality of figures must be improved

Based on the above comments, I would suggest publishing the manuscript after minor revision.

Author Response

Reply to reviewer’s comments

Journal:                   International Journal of Molecular Sciences 

Manuscript ID:       ijms-772595

Manuscript Title:    1,4‐Disubstituted 1H-1,2,3‐Triazoles for Renal Diseases: Studies of Viability, Anti-Inflammatory, and Antioxidant Activities

Authors:                  Ching-Yi Cheng*, Ashanul Haque*, Ming-Fa Hsieh*, Syed Imran Hassan, Serajul Haque Faizi, Necmi Dege and Muhammad S. Khan

We thank the editor and reviewers for their thorough review and for giving us an opportunity to revise our article. The comments/queries suggested by the editor and reviewers are replied and corrected in the revised manuscript (highlighted in red). We hope, the revised manuscript is suitable for publication in the International Journal of Molecular Sciences.

With best personal regards.

Sincerely yours,

Ching-Yi Cheng, Ph.D. (Corresponding Author)

Graduate Institute of Health Industry Technology, Research Center for Chinese Herbal Medicine and Research Center for Food and Cosmetic Safety, Chang Gung University of Science and Technology, No.261, Wenhua 1st Rd., Guishan Dist., Taoyuan City 33303, Taiwan.

Phone: 886-3-2118999, ext 5114

Fax: 886-3-2118866

Reviewer # 2

In this manuscript the authors present the extension of their research about the 1,4‐disubstituted 1H-1,2,3‐triazoles compounds. Also, the authors synthesized and characterized new compounds and demonstrate the biological activity. The compounds present an acceptable physicochemical, pharmacokinetic, and toxicity profiles and the anti-inflammatory effect and antioxidant activity was demonstrated in vivo. The experiment has been systematically performed and the manuscript is well organized therefore I suggest that this manuscript can be published on this journal after the correction of the following issues:

Comment 1:   The compounds F, X1 and X3 were better for preventing the increase of NO concentrations. Please explain why.

Reply: We thank the reviewer for this suggestion and accepting our article with minor revision. We have improved the discussion and included it in the revised MS.

See page: Page 6-7, subsection 2.2.2, lines 188-201.

Comment 2:   The quality of figures must be improved.

Reply: New improved Figures have been included in the revised MS and compressed image file.

Reviewer 3 Report

The authors studied 5 compounds in biological assays and performed docking analyses with two COX-2 and sPLA2. However, the analyses they have performed cannot put in relation to one another, as the authors did not study the effect of the compounds on the activity of these enzymes.

Moreover, a positive control is always missing!

Statistical analyses should be repeated. The controls are set either to 1 or 100%, but they should also show a standard deviation! This analysis does not fit with the statistical rules.

A discussion is missing and the conclusion is overestimated.

The authors wrote that this manuscript is a follow-up of their recent publication: Haque A et al: Synthesis, characterization, and pharmacological studies of ferrocene-1H-1, 2, 3-triazole hybrids. J. Mol. Struct. 2017, 1146; 536-545“. Unfortunately, this reference (No 9 in the manuscript) cannot be found. So, it is not clear what the difference between their recent publication and the present one is.

In more detail:

Abstract:

The authors wrote that „ most of the ferrocene-1H-1,2,3-triazole hybrids had limited antioxidant activity“. This sentence contrasts their conclusion that „ especially compounds X2-4, have the potential to be developed into anti-inflammatory and antioxidant agents“.

Results:

The authors used the MTT assay. However, this assay only studies effects on the metabolism and hence the viability of the cells. Cytotoxic effects cannot be studied. Please correct.

The authors should provide the data for HO-1 and SOD-2 in Supplementary Information.

NO and iNOS:

As the experiments have been performed with cells and not in vivo, the authors should replace „confirm“ by „indicate“: Inhibition of NO and iNOS indicated that ferrocene-1H-1,2,3-triazole hybrids could protect against inflammation in RMCs“.

How do the authors explain that X5 does not reduce TNFalpha induced iNOS gene expression (Fig. 4a), but reduces NO concentration (Fig. 4b)? Performance of the gene expression experiment on iNOS is not clearly described (Fig. 4a and c): „for 2 h at 24 h post incubation with TNF“? Does it mean that the cells were pretreated for 2 h with the compounds and subsequently stimulated with TNFalpha for 24h? Please rewrite!

The authors wrote: „As shown in Figure 4c, compounds F, X1, and X3 did not induce iNOS gene expression in RMCs. Overall, these results implied that compounds F, X1, and X3 154 were better at preventing the elevation of NO concentrations.“ Regarding results in Fig. 4b conclusion is not correct.

Results shown in Fig. 5a and 5b do not fit together concerning X3. X3 induces COX-2 in the western blot, while a reduction is shown for gene expression.

The numbers of independent experiments is missing in Fig. 5 and 6.

Docking results:

The authors showed in their biological assays that gene expression of MMP-9 and iNOS is reduced. Both are transcriptionally regulated by the transcription factor NFkappaB. The authors have undertaken docking analyses with the enzymes COX-2 and sPLA2s. Correlation of both experiments does not make any sense. To study the direct effect on these enzymes they have to perform assays which study whether the activity of these enzymes is reduced.

Materials and Methods

Replace cytotoxicity assay by viability assay

Round 2

Reviewer 3 Report

The authors have improved in some parts their manuscript. However, the most striking point has not been solved. „The authors studied 5 compounds in biological assays and performed docking analyses with two COX-2 and sPLA2. However, the analyses they have performed cannot put in relation to one another, as the authors did not study the effect of the compounds on the activity of these enzymes.“ The authors gave an explanation why they have undertaken their docking studies. Hoewever, these explanations are fare from being acceptable. The only way is to study the effect on both enzymes!

Further critical points:

  1. Now the authors have provided access to reference 4 (J. Mol. Struct, 2017). However, the authors have only studied the effect on LPS-induced NO. Therefore, it is highly overestimated that „X1-X5 show unique antiinflammatory activity in vitro models of LPS-induced inflammation.“
  2. Still the authors did not always correct that cytotoxicity cannot be studied in an MTT assay.
  3. The authors have now included the Fig. with SOD-2 and HO-1 in the Supporting Information part. They wrote „not all of the compounds induced antioxidant enzymes including HO-1 or SOD-2“. However, Fig. 1S shows no significant induction by all compounds!
  4. The authors wrote „in Figure 4c, compounds F and X1-X5 did not spontaneously induced iNOS gene expression in RMco.“ Looking at Fig. 4C, all compounds even inhibit iNOS expression without stimulation.

The next sentence does also not fit to Fig. 4a. Fig. 4a shows a significant inhibition except with X5. The authors wrote „only F, X1, and X3 significantly cause this inhibition.“

  1. The authors assigned COX-2 and cPLA1 as receptors. Both are enzymes.

Round 3

Reviewer 3 Report

Finally, the authors have presented cPLA2 and COX-2 activity assays.

By the way, no real expert in docking analyses will assign enzymes as receptors!!